# A Detailed Survey and Future Directions of Unmanned Aerial Vehicles (UAVs) with Potential Applications

Nourhan Elmeseiry [1], Nancy Alshaer [2] and Tawfik Ismail [1,3,*]

1. Wireless Intelligent Networks Center (WINC), Nile University, Giza 12677, Egypt; nahmed@niles.edu.eg
2. Department of EEC, Faculty of Engineering, Tanta University, Tanta 31527, Egypt; nancy.alshaer@niles.edu.eg
3. National Institute of Laser Enhanced Sciences, Cairo University, Giza 12613, Egypt
* Correspondence: tismail@cu.edu.eg; Tel.: +20-110-139-7777

**Abstract:** Recently, unmanned aerial vehicles (UAVs), also known as drones, have gained widespread interest in civilian and military applications, which has led to the development of novel UAVs that can perform various operations. UAVs are aircraft that can fly without the need of a human pilot onboard, meaning they can fly either autonomously or be remotely piloted. They can be equipped with multiple sensors, including cameras, inertial measurement units (IMUs), LiDAR, and GPS, to collect and transmit data in real time. Due to the demand for UAVs in various applications such as precision agriculture, search and rescue, wireless communications, and surveillance, several types of UAVs have been invented with different specifications for their size, weight, range and endurance, engine type, and configuration. Because of this variety, the design process and analysis are based on the type of UAV, with the availability of several control techniques that could be used to improve the flight of the UAV in order to avoid obstacles and potential collisions, as well as find the shortest path to save the battery life with the support of optimization techniques. However, UAVs face several challenges in order to fly smoothly, including collision avoidance, battery life, and intruders. This review paper presents UAVs' classification, control applications, and future directions in industry and research interest. For the design process, fabrication, and analysis, various control approaches are discussed in detail. Furthermore, the challenges for UAVs, including battery charging, collision avoidance, and security, are also presented and discussed.

**Keywords:** UAV; drone; review; control; design; applications; future research trends

## 1. Introduction

Unmanned aerial vehicles (UAVs) have become increasingly important in different disciplines of civilian and military applications due to their improved endurance and stability in several conditions and operations [1]. According to the Association for Unmanned Vehicle Systems International (AUVSI), there were more than 2900 UAVs across more than 900 companies providing services around the world in 2020 [2]. A UAV can be described as a pilotless aircraft that can fly and stay airborne without a human operator onboard, able to perform critical operations without risking a human's safety, and operating more cost effectively than equivalent manned systems [3]. They can be either remotely piloted, whereby the control actions are performed from a ground control station using a remote control, or operated autonomously, whereby the UAV is capable of performing the control actions onboard using autopilot and various sensors, including IMUs and GPS [4,5]. They can perform a variety of functions, including remote sensing, transportation, armed attacks, and search and rescue missions. UAVs can be referred to using different terms, such as drones, remotely piloted aircraft (RPA), unmanned aerial systems (UASs), and remotely piloted aerial systems (RPASs). These various terms are often due to the different criteria and requirements of military and civilian systems. Usually, in some studies from the Department of Defense (DoD), they are referred to as UASs, while the term "drone" is

quite popular in the media and commercial companies [6]. In this study, we refer to these systems as UAVs.

Initally, UAVs were mainly used for military applications such as reconnaissance, surveillance, and weapon delivery. Before the first manned airplane flight, primitive UAV technology was used in military applications such as combat and surveillance in at least two wars [7]. In 1917, Dr. Peter Cooper and Elmer A. Sperry invented the automatic gyroscopic stabilizer, which helped the aircraft to fly straight and maintain the flight level. Their technology has been used to convert a U.S. Navy Curtiss N-9 trainer into the first radio-controlled UAV, as shown in Figure 1. It was a pilotless biplane made out of wood [8]. The Sperry Aerial Torpedo flew 50 mi carrying a 300 lbs bomb in several tests.

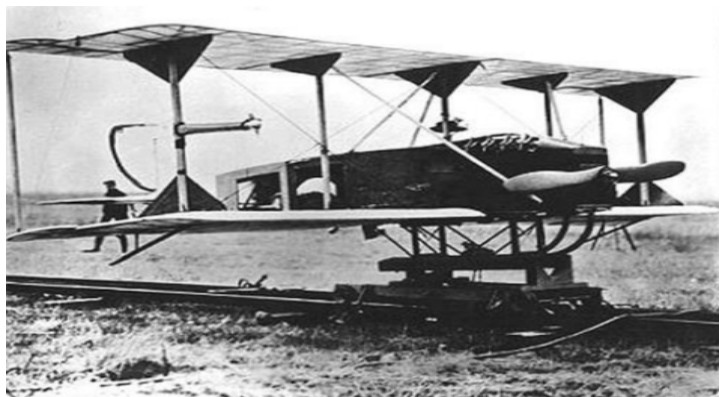

**Figure 1.** The first unmanned aerial vehicle: "Sperry Aerial Torpedo" [7].

Gradually, UAVs have been employed in several civil applications, including agriculture, remote sensing, aerial photography, and delivery. Various e-commerce companies, such as Amazon, are starting to adopt UAVs for their delivery services. Recently, UAVs have been used in the wireless communication field to provide network coverage and improve network connection in isolated areas [9]. Regarding the COVID-19 pandemic, UAVs have been used to monitor civilian movements and social gatherings to reduce the risk of spreading the disease and report gatherings and lockdown violations to reduce the possibility of police and health authorities being infected [10].

An analysis carried out in the USA in 2014–2017 showed that the main applications of UAVs are in photography, inspection and maintenance, mapping, surveillance and monitoring, and precision agriculture [11]. The development of micro-electromechanical systems (MEMSs), sensors, fabrication, navigation methods, remote control capabilities, and power systems have enabled the design and manufacture of a wide variety UAVs, which can be used in many circumstances where it is either difficult or dangerous for human beings to be involved [12]. Therefore, UAVs vary widely in size, configuration, and performance characteristics in order to carry several payloads, such as cameras, sensors, navigation equipment, and communication equipment, depending on the flight mission. Therefore, there are different classifications of UAVs according to various parameters, including their size, range, weight, engine type, and configuration.

The rest of this review is organized as follows: In Section 2, the different classifications of UAVs are discussed. The mechanical design processes of UAVs and the different analyses performed with software and in the lab are presented in Section 3. In Section 4, various navigation and control approaches that are used for UAVs are given. Existing applications for UAVs in various fields are explored in Section 5. The key challenges, limitations, and recommendations are presented in Section 6. Finally, proposed future research directions and conclusions are given in Sections 7 and 8, respectively.

## 2. Classification of UAVs

There has been a significant effort to develop and deploy UAVs for particular purposes in recent years. These efforts have enabled the design and fabrication of different types of

UAVs with different performance characteristics. Therefore, the classification of UAVs is required to present how widely UAV systems vary and to demonstrate their capabilities. This section discusses the different UAV classifications that have been proposed according to various parameters, including their size, weight, range and endurance, maximum altitude, engine type, and configuration, in order to assist in selecting the appropriate parameters for UAVs.

### 2.1. Size-Based Classification

One of the most effective metrics for categorizing UAVs is according to their size. UAVs come in different sizes to serve a variety of purposes. They range in scale from insect-sized devices at one extreme to large aircraft sizes at the other. Ultra-small UAVs can be carried by a single person. They may also be the size of a bird or a large insect, according to [13,14]. They are designed to fly inside buildings and areas inaccessible to humans to monitor these places using miniature sensors implemented in the UAV. Larger UAVs can fly for long periods of time at high altitudes within a wide range far from the base station to perform various missions, such as field surveillance and penetrating attacks. In the military, they are used to carry weapon payloads over long distances to their destination [13,15]. Accordingly, UAVs could be classified into four categories based on their sizes, as follows:

1. "Ultra-small UAVs", which includes NAVs (Nano Aerial Vehicles) with a maximum wingspan length less than 7.5 cm [16]—Robobee X-wing is an example of an NAV with a very small wingspan of 3.5 cm, shown in Figure 2a [17]—and MAVs (Micro Aerial Vehicles) with a length between 7.5 and 15 cm [16]; DelFly Micro is an example of a MAV with a wingspan of 10 cm, shown in Figure 2b [18].
2. "Small UAVs", also known as mini-UAVs, with a maximum dimension between 15 and 200 cm [19]; WASP AE is an example of a small UAV; it is a fixed-wing hand-launched UAV that has a wingspan of 100 cm and a length of 76 cm, shown in Figure 2c.
3. "Medium UAVs", which applies to UAVs with a wingspan between 2 and 10 m; NASA SIERRA is an example, with a wingspan of 6.1 m and length of 3.9 m, shown in Figure 2d.
4. "Large UAVs" are used to provide long endurance flights for surveillance and carry weapons to desired areas with a length greater than 10 m. An example is Global Hawk, with a length of 13.4 m and a wingspan of 35.35 m, shown in Figure 2e [20].

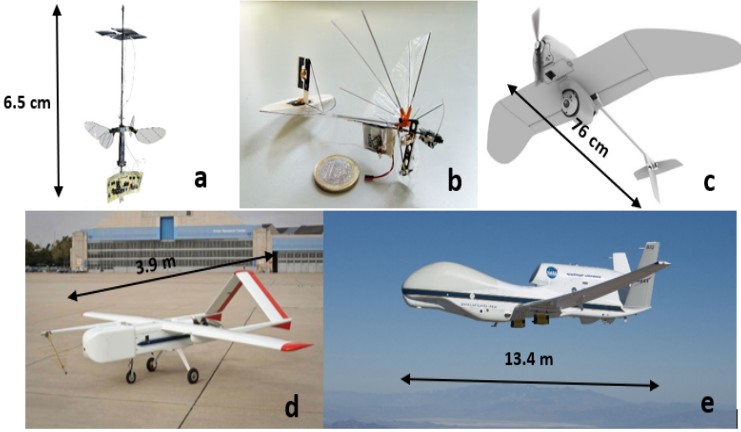

**Figure 2.** Different UAV classes based on size: (**a**) ultra-small: NAV; (**b**) very small: MAV; (**c**) small UAV; (**d**) medium UAV; (**e**) large UAV.

A summary of this classification is shown in Table 1.

**Table 1.** UAV classification based on size.

| Class | Maximum Dimension |
| --- | --- |
| Ultra-Small (NAV) | <7.5 cm |
| Very Small (MAV) | 7.5–15 cm |
| Small | 15–200 cm |
| Medium | 2–10 m |
| Large | >10 m |

*2.2. Range, Endurance, and Altitude*

In the analysis of UAV performance, range is considered one of the most important attributes. It can be calculated easily using different parameters. Range is dependent on other UAV parameters, especially the weight of the payload. It identifies how far a UAV can fly away from its ground control station. Another important metric is the endurance of the UAV, which describes the time that a UAV can fly before recharging/refueling. In other words, it describes how long a UAV can fly. The endurance of UAVs can vary from 1 to more than 36 h, according to the mission. The volume and mass of the fuel or battery load affect the endurance and efficiency of the UAV. The mass of the fuel can range between 10% and 50% of the UAV, which directly affects the endurance of the UAV. Therefore, the amount of fuel should correspond with the endurance required for the UAV [21]. The range and endurance parameters are related to each other as the longer a UAV can fly, the larger its range will be. Therefore, different classifications based on the range and endurance of the UAVs were proposed. The US military developed a classification of UAVs based on range and endurance. They classified UAVs into five categories, as follows:

1. "Very close-range UAVs" are UAVs with a maximum range of 5 km, usually used by the Marine Corps and the army as a model air vehicle.
2. "Tethered UASs" provide power and communication to the UAV through a permanent physical link in the form of a flexible wire or cable. They employ quadcopters or other multicopter UAVs so it can hover. These systems are utilized when the necessary flight endurance is more significant than a free-flying UAV and a small operating area is required. The Orion 2 UAS [22] provides completely automated, continuous airborne surveillance over broad regions during the day and night. It has an endurance length of 24 h per day, which means it can stay in the air indefinitely at altitudes up to 100 m, covering extended ranges to up to 10 km.
3. "Close-range UAVs" have a maximum range of 50 km and endurance lengths of 1–6 h depending on the mission; they are mainly required for reconnaissance and surveillance missions.
4. "Short-range UAVs" have a maximum range of 150 km and endurance lengths of 8–12 h. They also are required for reconnaissance and surveillance missions.
5. "Mid-range UAVs" have a maximum range of 650 km. They are usually required to be ground or air-launched for reconnaissance and surveillance work and the collection of meteorological data.
6. "Endurance UAVs" are UAVs that have an endurance of more than 36 h, have a maximum range of 300 km, and can operate from the ground or sea [13].

The five classification categories are summarized in Table 2.

The maximum altitude of a UAV is another important performance metric in classifying UAVs. Due to the requirement of low visibility, high-altitude UAVs are used for military purposes to avoid being detected and for reconnaissance and surveillance missions to cover wide areas. Therefore, UAVs can be classified into three classes according to their maximum altitude: "low altitude" for altitudes less than 1000 m, "medium altitude" for altitudes between 1000 m and 10,000 m, and "high altitude" for altitudes more than 10,000 m [23]. Other classifications consider more than one parameter. A classification based on endurance and altitude classified UAVs into seven categories: "MAV and NAV," as discussed previously. They are used because of their sizes; they can fit in individual sol-

diers' backpacks. They usually operate at very low altitudes (<330 m) and short endurance lengths of 5–30 min. "VTOL UAVs" do not require a runway. Therefore, they are usually used in rough terrains. They can fly at varying altitudes based on the mission profile but usually fly at low altitudes. Hovering flights limit their endurance due to high power consumption. "LASE", low-altitude short-endurance UAVs, as with VTOL UAVs, also do not require runways. They have an endurance length of 1–2 h."LASE Close" UAVs can take off and land vertically. They have a larger size and weight than LASE UAVs. These UAVs can fly to a maximum altitude of 1500 m and remain airborne for multiple hours. "LALE", low-altitude long-endurance UAVs, can carry payloads of several kilograms at altitudes of a few kilometers for extended periods. "MALE", medium-altitude long-endurance UAVs, are larger than LALE UAVs. They can fly at altitudes up to 9 km. "HALE", high-altitude long-endurance UAVs, are the largest UAVs. They can fly at altitudes more than 20,000 m and have an endurance of over 36 h [24].

**Table 2.** UAV classification based on range and endurance.

| Category | Maximum Range | Maximum Endurance |
|---|---|---|
| Very Close Range | 5 km | <6 h |
| Tethered UAS | 10 km | 24 h |
| Close Range | 50 km | 6 h |
| Short Range | 150 km | 12 h |
| Mid-Range | 650 km | 12–36 h |
| Endurance | 300 km | 36 h |

*2.3. Weight-Based Classification*

The weight of a UAV corresponds to the predicted kinetic energy produced during impact, which is considered to be the main factor that affects the safety of missions [25]. Therefore, some researchers and organizations have classified UAVs into ten classes based on their weight: "nano UAVs", which weigh less than 200 g, "micro UAVs", which weigh between 200 g and 2 kg, "mini UAVs", which weigh between 2 kg and 20 kg, "light UAVs" which have weight ranges between 20 kg and 50 kg, "small UAVs", with weights between 50 kg and 150 kg, "tactical UAVs" which weigh between 150 kg and 600 kg, "MALE UAVs" and "HALE UAVs", which have weight ranges between 600 kg and 1000 kg, "heavy UAVs", that weigh between 1000 kg and 2000 kg, and "super heavy UAVs" which have the largest weight between 2000 kg and 24,950 kg. Ravn X is considered to be the world's heaviest UAV, which weighs 24,950 kg [26,27]. This classification is represented in Table 3.

**Table 3.** UAV classification based on weight and range.

| Class | Maximum Weight | Maximum Range |
|---|---|---|
| Nano | 200 g | 5 km |
| Micro | 2 kg | 25 km |
| Mini | 20 kg | 40 km |
| Light | 50 kg | 70 km |
| Small | 150 kg | 150 km |
| Tactical | 600 kg | 150 km |
| MALE | 1000 kg | 200 km |
| HALE | 1000 kg | 250 km |
| Heavy | 2000 kg | 1000 km |
| Super heavy | 24,950 kg | 1500 km |

Furthermore, the U.S. Department of Defense categorized UAVs into five groups based on the maximum take-off weight (MTOW), altitude, and speed.

- "Group 1" are hand-launched and portable UAVs. Their missions are reconnaissance, surveillance, and target acquisition. They are lightweight UAVs with MTOW less than 20 pounds and low altitudes less than 396.24 m above ground level (AGL).
- "Group 2" are medium-sized UAVs that can be launched using a catapult, mainly used for reconnaissance, surveillance, and target acquisition. They have MTOW between 20 pounds to 55 pounds, operating at altitudes less than 1066.8 m above ground level. They can carry heavier payloads than "Group 1", which affects their endurance.
- "Group 3" are larger UAVs than those in "Group 1" and "Group 2"; the majority of these UAVs are used to carry weapons. They do not require an improved runway. Therefore, they are usually used in rough terrains. Their MTOW is less than 1320 pounds, operating at medium altitudes less than 5486.4 m mean sea level (MSL).
- "Group 4" operates at the same altitude as "Group 3". They are larger than the previous groups. They have MTOW greater than 1320 pounds. They can carry heavier payloads, but they may require improved runways.
- "Group 5" are the largest UAVs, with MTOW greater than 1320 pounds and an altitude of over 5486.4 m mean sea level. They have long range and endurance parameters that allow them to carry out advanced operations, such as wide-area surveillance and penetrating attacks. However, they require enhanced areas for launch and recovery [15].

This categorization is represented in Table 4.

**Table 4.** UAV DoD categorization [15].

| Category | MTOW | Altitude (m) | Airspeed |
|----------|------|--------------|----------|
| Group 1 | >20 pounds | <365.76 AGL | <100 knots |
| Group 2 | 21–55 pounds | <1066.8 AGL | <250 knots |
| Group 3 | <1320 pounds | <5486.4 MSL | <250 knots |
| Group 4 | >1320 pounds | <5486.4 MSL | Any airspeed |
| Group 5 | >1320 pounds | >5486.4 MSL | Any airspeed |

*2.4. Engine Type-Based Classification*

As investigated earlier, UAVs come in different sizes, weights, and other varying performance characteristics, which require different engine types to operate the UAV properly, depending on these characteristics, to achieve the desired operation. For example, as the weight of the UAV increases, the size of the engine also increases. Additionally, the engine type can affect the endurance and range of the UAV [13]. Engines can be classified into two main categories: "chemical engines" and "electrical engines." Piston and electric engines are the most commonly used types of engines. Electric engines are generally used for light and small UAVs, while piston engines are used with large UAVs that are required to carry heavy payloads [28,29].

Piston engines can be divided into four types: "two-stroke engines", which are highly used in UAVs due to their low weight, and they have high power to weight ratio, but they have a high level of noise compared with other types of engines such as a "Wankel engine" and a "four-stroke engine." "Four-stroke engines" are used for heavy UAVs, and they have a relatively higher efficiency than two-stroke engines, but they are heavier and have a very low power to weight ratio, which makes them less used for UAVs. "Wankel engines" have high power to weight ratios and simple constructions. However, they have high manufacturing costs and high emission levels of carbon dioxide and "rotary engines." Finally, "jet propulsion engines" have very high power to weight ratios due to their small weight. They can be categorized into two types: the "turbojet engine" is one of the simplest turbine engines, it has a very high power to weight ratio due to its light weight, but it consumes a large amount of fuel. Another type is the "turboprop engine." The other engine category is "electric engines." They can generate a proportional torque to the applied supply voltage, which makes them more efficient than other engines, with a

light weight, which makes them appropriate for small UAVs. However, they have limited endurance relative to the power supply. They are usually operated using batteries, although they also can use solar cells such as: "lithium–sulfur batteries, Li-S", "lithium–ion cells, Li-Po", "nickel–metal hydride cells, Ni-MH", "hydrogen fuel cells", and "photovoltaic modules" [30].

### 2.5. Configuration-Based Classification

UAVs vary widely in their configurations depending on their missions. However, they can be classified based on their configuration into four classes: horizontal take-off and landing (HTOL), vertical take-off and landing (VTOL), hybrid UAVs, and unconventional UAVs. Each one of these classes is discussed in detail.

#### 2.5.1. Horizontal Take-Off and Landing (HTOL) UAVs

These UAVs are required to accelerate horizontally along a runway in order to take off and land [31]. There are conventional fixed wings in the airplane's body, and they can only take off and land horizontally, as shown in Figure 3a. They can be categorized into four different classes: tailplane-aft, tailplane forward, tail-aft on booms, and tailless or flying wing UAVs. Their propulsion systems can be at the rear of the fuselage or at the front side [14]. They consist of rigid wings with an airfoil, which allows them to fly based on the lift generated by the forward airspeed. Thus, they support long-range and endurance flights with high speeds. In addition, these UAVs can carry heavier payloads and consume less energy than rotary-wings [32]. However, they need a runway to take off and land, which makes them restricted for improved areas and means they cannot perform hovering tasks since they need to constantly move forward during missions [33].

#### 2.5.2. Vertical Take-Off and Landing (VTOL) UAVs

These are UAVs that have the ability to take off and land vertically. They do not require a runway for taking off or landing, as shown in Figure 3b. They are rotary-wing UAVs with high maneuverability advantages, using rotary blades instead of the fixed wing as the flying mechanism. They can perform low-altitude flights and hovering tasks. The rotary blades produce a thrust force that should be higher than the weight of the UAV in order to generate lift.

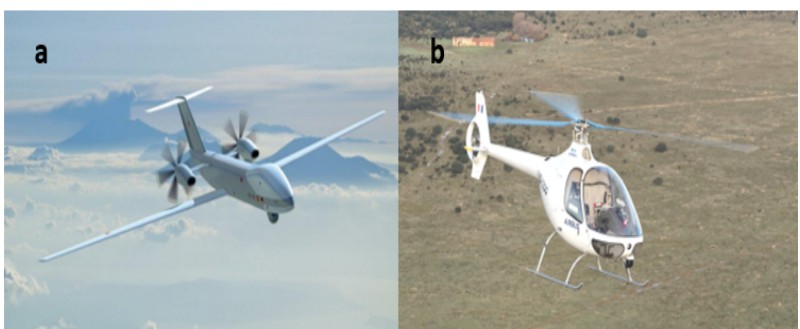

**Figure 3.** (**a**) Fixed-wing HTOL UAV (European MALE RPAS) [34]; (**b**) helicopter VTOL [35].

They have different configurations depending on the number of the rotors, such as two-rotor "helicopters", where the main rotor is used for navigation while the other is used to generate counter torque and control the heading. The gas engine enables longer-endurance flights compared with multi-rotor UAVs. They can carry heavy payloads to perform different tasks. However, they have a complex mechanical mechanism and high costs [33]. Other single-rotor UAVs include the co-axial rotor and tandem rotor.

Multi-rotor VTOL UAVs also can hover and have high maneuverability. They can fly in a horizontal or vertical manner and can hover at a fixed height. They have different configurations, shown in Figure 4, such as: (1) monocopters, which have a single rotor/blade for flight [36,37]; (2) twincopters, which have two rotors/blades for flight and the two

rotors rotate opposite to each other to generate thrust [38]; (3) tricopters, which have three blades/rotors, can perform flight and hovering in all directions, and have two different configurations, "Y" and "T" [39]; (4) quadcopters, which are rotary-wing UAVs that have four blades/rotors, and have different configurations that are based on the position of the rotors concerning the frame/body coordinate system [40]; (5) pentacopters, which have five blades/rotors and can carry heavier payloads than quadcopters [41]; (6) hexacopters, which have six blades/rotors, six degrees of freedom, and have two configurations, "X" and "Plus". They can carry heavier payloads than pentacopters [42]; (7) octocopters, which have eight blades/rotors [43]; (8) decacopters, which have ten blades/rotors; and finally, (9) dodecacopters, which use twelve blades/rotors for flight and have the largest weight and can carry heavier payloads than any other multi-rotor [44]. They can fly at different altitudes and have a simple mechanical structure compared with the other configurations. However, they have limited endurance and low speed compared with fixed-wing UAVs.

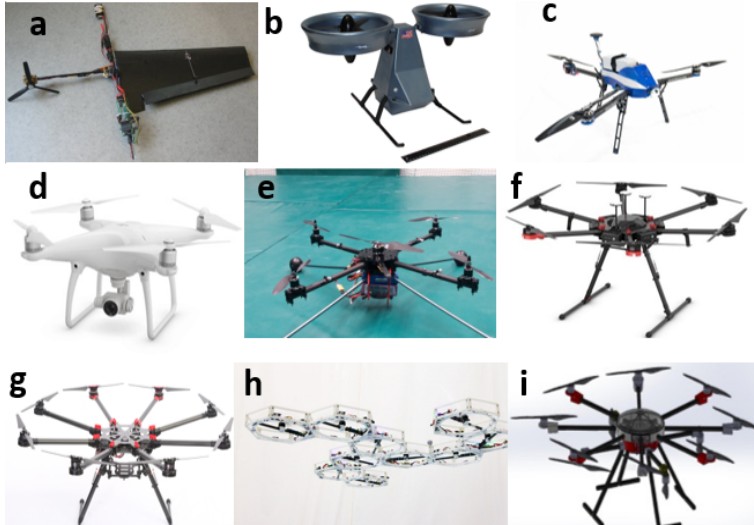

**Figure 4.** Multirotor UAVs: (**a**) monocopter [45], (**b**) twincopter [46], (**c**) tricopter [47], (**d**) quadcopter [48], (**e**) pentacopter [49], (**f**) hexacopter [50], (**g**) octocopter [51], (**h**) decacopter [52], (**i**) dodecacopter [44].

### 2.5.3. Hybrid UAVs

Recently, there has been massive interest in combining the benefits of HTOL and VTOL configurations by designing UAVs capable of VTOL and long endurance flights by changing their configuration into a fixed wing during the mission, which allows the speed to increase and less power to be consumed. Hybrid UAVs are categorized, as shown in Figure 5, into "tilt-rotor", "tilt-wing", "tilt-body", "ducted fan", and "tail-sitter" UAVs. Tilt-rotor UAVs have moving rotors that are vertical during take-off, landing, and vertical flight. However, they can tilt forward 90 degrees to be positioned horizontally for horizontal flight [53]. The tilt-wing configuration changes the degree of the whole wing from 0 to 90 degrees for VTOL and vertical flights. However, they tilt the wing instead of motors [54]. Tilt-body UAVs can change incidence angle by rotating about the spanwise shaft. A ducted fan is a type of UAV where the thrusters are enclosed within a duct [55]. Finally, a tail-sitter is a UAV that takes off and lands vertically on its tail. It tilts forward for horizontal flight and changes the fuselage orientation after take-off. An example is the Northrop Grumman Tern and MARS Electric Reusable Flyer.

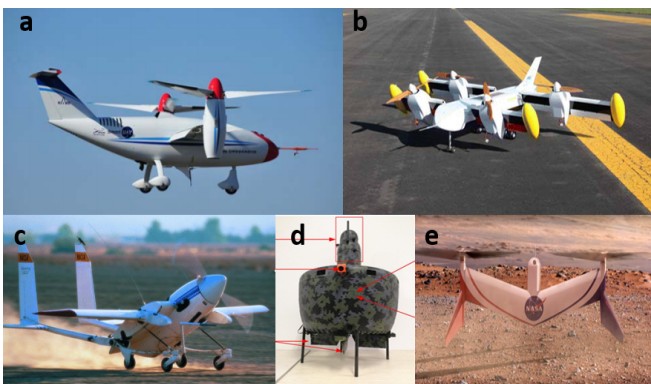

**Figure 5.** Hybrid UAVs: (**a**) tilt-rotor [56], (**b**) tilt-wing [57], (**c**) tilt-body [58], (**d**) ducted fan [55], (**e**) tail-sitter [59].

2.5.4. Unconventional UAVs

Unconventional UAVs are UAVs that have unique structures that cannot fit in any of the previous categories mentioned above, such as "bio UAVs," which are used in military and civilian applications, inspired by insects and birds, with small sizes and weights to serve several applications with the help of miniaturization technologies. Several types of unconventional UAVs are presented in Figure 6. "Live bio UAVs" provide the capability to control birds' and insects' flights. For example, researchers attached an electronic chip to a pigeon's brain to control the pigeon's movements remotely using very thin electrodes that were implanted into the pigeon's brain in locations responsible for movement. "Taxidermy bio-UAVs" use the dead bodies of birds or animals as the structure of the flying platform, such as Orvillecopter, a cat body quadcopter. "Bio-inspired UAVs", such as "flapping wing UAVs", consist of flexible, flapping wings that use an actuation gear mechanism for their flapping motion. Flapping wing UAVs have flexible and lightweight wings, which affect their stability and aerodynamic proficiency. They are considered to have more complex mechanisms than fixed-wing and rotary-wing UAVs due to their aerodynamics complexity. Festo Smart bird is an example of a flapping wing UAV.

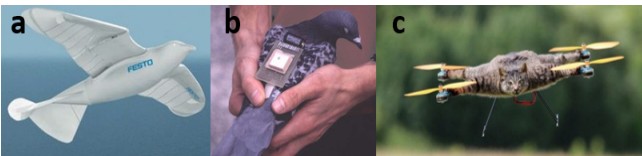

**Figure 6.** Unconventional UAVs: (**a**) flapping wing [60], (**b**) live bio UAV [61], (**c**) taxidermy bio UAV [62].

## 3. Mechanical Design and Analysis of UAVs

Designing a UAV is an iterative process that requires extensive planning. As discussed earlier, UAVs have different configurations. However, the design procedure of a UAV consists of three main steps, which are the conceptual design, preliminary design, and detailed design, regardless of the UAV configuration, size, and defined mission. In this section, the design and analysis processes for UAVs are discussed in detail.

### 3.1. Design and Analysis of a UAV

Designing a UAV is an iterative process, as shown in Figure 7. It mainly involves three phases, as follows:

1. "Conceptual design", in which the UAV is designed in the concept of brainstorming ideas and analyzing the advantages and disadvantages of each idea when it comes to implementation without any specific calculations. The design parameters are determined according to the decision-making process and selection technique. A

potential design should be presented according to the strengths and weaknesses of each design idea regarding the pre-determined design parameters [21,63].

2.  "Preliminary design" involves the sketches of the design that should be the outcome of some calculation procedures. Therefore, the parameters determined in this phase are still not final, and they should be adjusted according to the analysis applied and design alteration. However, this step is crucial as the outcome parameters of this step will be used in the detailed design final phase. Therefore, the results of the preliminary design should be accurate.

3.  "Detailed design" is the final phase in the design process. It involves a detailed overview of every design aspect, including detailed CAD models, simulations, and final characteristics.

Choosing materials suitable for UAV manufacturing is a big concern. Different materials can be used for manufacturing a UAV, including wood, aluminum, fiberglass, ABS, and carbon fiber. The great concern is that the material should be lightweight with good mechanical properties and high strength, which is a trade-off. The designer should choose the most suitable material based on the design specifications [64].

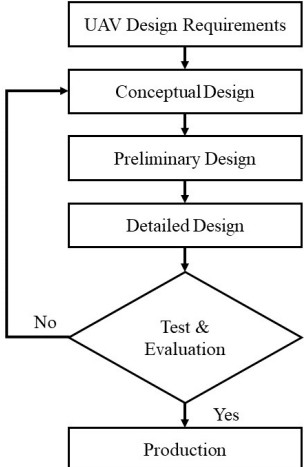

**Figure 7.** UAV design process.

### 3.1.1. Designing a Fixed-Wing UAV

Fixed-wing UAVs are the most developed due to the ease of their design and fabrication. However, the design of fixed-wing UAVs is based on trial and error methods which increase the cost of the design process and consume time. Therefore, several researchers developed various design methodologies to design fixed-wing UAVs. The methodologies start with estimating the total weight and then determining the wing area and engine thrust. Then, the preliminary design and evaluation of the results are performed with the help of analysis methods such as computational fluid dynamics (CFD) [65].

Regarding the sizing methodology of a fixed-wing fueled UAV, the sum of the weights of the various components of the UAV must equal the total weight of the UAV. This relationship is represented in the following equation.

$$W_T = W_{airframe} + W_{payload} + W_{engine} + W_{fmission} \tag{1}$$

When divided by the wing area, the equation expresses the equality between the wing loading of the UAV and the sum of the portions of that wing loading due to each of the components:

$$\frac{W_T}{S} = \frac{W_{airframe}}{S} + \frac{W_{payload}}{S} + \frac{W_{engine}}{S} + \frac{W_{fmission}}{S} \tag{2}$$

The overall wing loading, indicated on the left side of the equation, is usually set to satisfy the customer's performance requirements. This is known as constraint analysis. The client sets the payload, and the airframe and engine wing loading sections are calculated using weight analysis. The wing loading component for mission fuel use is computed using mission analysis. The technique for applying this equation to design and optimize the aircraft entails selecting values for all terms except the term of payload wing loading portion and then calculated the wing reference platform needed using the weight of the required payload.

$$S = \frac{W_{payload}}{\frac{W_T}{S} - \frac{W_{airframe}}{S} - \frac{W_{engine}}{S} - \frac{W_{fmission}}{S}} \tag{3}$$

The fundamental difference between a fueled and an electric-powered UAV is how the weight varies during the flight. The fuel in a fueled UAV is consumed during the trip. Therefore, the weight of the UAV lowers as the mission progresses. The weight of the battery-powered UAV, on the other hand, remains constant unless the payload is dumped or the mission is terminated. Furthermore, the amount of energy recovered from a given quantity of fuel is now significantly greater than that extracted from a similar mass of batteries; hence, fueled UAVs are frequently the least costly means to achieve lengthy flights.

Regarding the sizing methodology for electric-powered UAVs, the sum of the weights of the various components of the UAV must equal the UAV's total weight. The proportionality between the wing loading of the UAV and the sum of the portions of that wing loading due to each of the components is expressed by dividing that equation by the wing area, as presented in the following equation:

$$\frac{W_{takeoff}}{S} = \frac{W_{airframe}}{S} + \frac{W_{payload}}{S} + \frac{W_{battery}}{S} + \frac{W_{motor\&prop}}{S} \tag{4}$$

The same strategy used in the fueled UAV for sizing and optimization is used for an electric UAV, which is represented in the following equation:

$$S = \frac{W_{payload}}{\frac{W_T}{S} - \frac{W_{airframe}}{S} - \frac{W_{battery}}{S} - \frac{W_{motor}}{S}} \tag{5}$$

### 3.1.2. Designing a Rotary-Wing UAV

Designing a multi-copter rotary-wing UAV starts with estimating the total weight of the UAV with all its equipment. According to this weight, the optimum design values of components are calculated or selected. The design process of a multi-copter has the following flow: first, the total mass of the vehicle is determined; second, the appropriate number and power of the motors is selected, and the hardware ESC suitable for the selected motors is determined. After that, the propeller size that would be the best for the mass and motors' power is selected. Then, the battery needed to power the vehicle is determined. In multi-copters, the battery weight can affect the overall weight, so redesigning could be considered if the weight exceeds the expected weight. After reaching stable selections from the above procedures, the size of the frame that fits all the requirements and does not exceed the expected total weight is determined. After that, we can determine the controller and sensors used and any other additional equipment and distribute the weight over the frame structure. This stage may require re-determination of the size of the frame to carry all the required equipment. The pre-manufacturing steps are analyzing the loads produced due to different sensors and equipment such as stresses and deformations, selecting the material that will sustain the existing stresses if there is no suitable material available, then redesigning as required. Another stage in the pre-manufacturing is software development to simulate the control applied in real conditions and ensure the stability of the multi-copter,

and finally, manufacturing and testing the multirotor in real conditions to ensure that the design process is successful [44].

T is the thrust generated by each rotor, which can be calculated using the following Equation [66].

$$T = 0.28 \times P \times D^3 \times N^2 \times 10^{-10} \text{ (N)} \tag{6}$$

where $P$ is the pitch of the propeller, $D$ is the diameter of the propeller, and $N$ is the speed of the motor in rpm.

### 3.2. Analysis of a UAV

3.2.1. Structural Analysis

Structural analysis is one of the essential applications of the finite element method (FEM). It is performed by first determining the loads acting on the UAV frame, including pressure loads and reaction forces, then applying the boundary conditions on the frame at the acting points/areas. The use of structural analysis confirms whether the design of the UAV with a specific material can withstand the required payload in different environmental conditions or not.

In [66,67], structural analysis of a quadcopter UAV was carried out in order to test the material strength for the payload.

3.2.2. Vibration Analysis

Vibration can be an undesired side effect of poor product design or due to the environment conditions in which the product is operating. It can significantly impact the durability and fatigue of the design, which may lead to a shorter service life. Vibration analysis starts with a modal analysis to find the mode shapes and the corresponding natural frequencies. Then, with the help of analysis software such as ANSYS, the corresponding deformation can be computed [68].

In [69], vibration analysis of a quadcopter frame was analyzed, and the first six mode shapes with their corresponding natural frequencies were obtained.

3.2.3. Computational Fluid Dynamics Analysis

Computational fluid dynamics (CFD) is a branch of fluid mechanics that analyzes problems involving fluid flows using numerical methods and algorithms. The CFD analysis is divided into three stages: the first step is pre-processing, which includes the generation of geometry, flow conditions, and thermal parameters. The second stage includes numerical analysis or the generation of numerical solutions. The final stage includes post-processing analysis, visualization of the solution, and the generation of final reports that include all information about the CFD analysis. Lift, drag, lift coefficient (CL), and drag coefficient (CD) are all measured and examined separately in this technique [70].

## 4. Control of UAVS

UAV control encompasses a wide range of treatments and technologies. Due to the unstable nature of UAV configurations, the design and analysis of UAV control must follow an accurate approach [4]. Therefore, the mission of the flight control system of a UAV is critical to ensure the stable behavior and the desired performance of the UAV, which could be affected by external disturbances. Figure 8 represents the architecture of the UAV control system. The controller aims to minimize the error between the desired state and the estimated state, which could be the position, velocity, or attitude of the UAV with disturbances and operations in different environmental conditions.

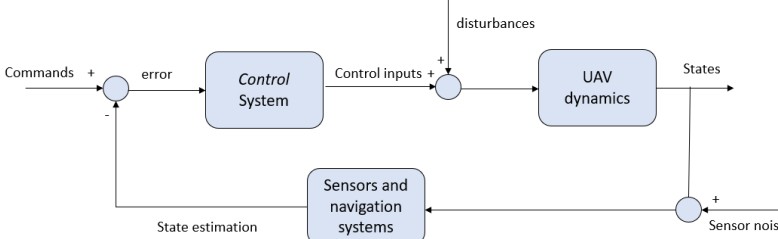

**Figure 8.** UAV control system architecture.

The flight controller consists of three main parts. The first is the kernel control law used to ensure the asymptotic stability of the UAV's motion within the air. The second part is the command generator required to generate references for the kernel control, and the third part is the flight schedule to generate a set of flight arrangements to achieve the required mission [71]. Controlling the flight of a UAV can be achieved by controlling the altitude to keep the UAV at the required height from the ground level and controlling velocity and orientation to achieve and follow the specified paths [72]. Different control laws can be used for UAVs. Linear controllers, such as PID, PD, LQR, and H∞ and non-linear controllers, including Backstepping, sliding mode, nested saturation, fuzzy control, gain-scheduling, and NDI can be used. Linear controllers were used in the early years to achieve autonomous flights, including:

- A Proportional–Integral–Derivative PID controller, which is one of the most used controllers for UAVs. The controller gains are selected to tune the current state of the UAV to a reference state. Due to their ease of use, they are quite popular for UAVs. However, they have limitations in robustness and optimality [72].
- A Linear Quadratic Regulator LQR controller works in dynamic systems to minimize a suitable cost function. It is considered robust concerning process uncertainty with a substantial stability margin to errors in the loop. However, it requires the full state of the system, which not always possible.
- The H ∞ Loop-Forming Method, which integrates robust control with classical loop forming control, has high robustness compared to other methods. However, it can be inefficient for large-scale UAV control.

Non-linear controllers are used to deal with system non-linearity and coupling components of UAV state variables.

- The Sliding Mode Controller (SMC) is based on Lyapunov stability principles. It forces the system to slide along a recommended path using a discontinuous control signal. It is insensitive to disturbances, modeling errors, and parametric uncertainties.
- The Backstepping Controller partitions the controller task into several steps to process and gradually stabilize each sub-system. It has a fast convergence of the algorithm, which consumes lower computational resources and deals with external uncertainty.
- Robust Control Algorithms are used to ensure the effectiveness of the controller performance within the acceptable disturbance range.

In [73], the brain–computer interface remote control of UAVs was reviewed, in which the brain signals (e.g., EEG signals) were converted into control commands for the UAV, and the feedback information of the system was carried to the operator. This system included data acquisition, prepossessing, feature extraction, and classification.

Recently, learning-based control methods have become popular for UAV control. The learning techniques can adapt to the environmental issue unlike the traditional controllers and learn from real-world data and behave accordingly. Furthermore, the advances in machine learning and deep learning have enabled the control of the UAV to recognize a pattern, follow a path on a map, or track an object autonomously and with high accuracy.

Trajectory planning is an important technology in UAVs; it refers to the action of planning an optimal path for the flight of the UAV, which is defined between a starting point and

an ending point, taking into consideration the fuel/energy consumption, mission time, the traveled area and the maneuverability [74]. Various algorithms were developed to achieve trajectory planning of a UAV, including but not limited to the A* algorithm [75,76], particle swarm optimization algorithm [77], neural networks [78], and the genetic algorithm [79] .

### 4.1. Fixed-Wing UAV Controllers

In [80], six control techniques were applied to control the flight of a fixed-wing UAV. Two linear controllers are the Proportional–Integral–Derivative (PID) controller and Proportional-Derivative (PD) controller, and four non-linear controllers are Backstepping, sliding mode, nested saturation, and fuzzy control. Each controller was used to control the movements of the fixed-wing UAV in the three main movements of altitude, yaw, and roll. The mathematical model of the fixed-wing UAV was derived, and each control signal for each movement for the six controllers was represented. The results showed that the sliding mode and Backstepping controllers had an acceptable response with regard to the altitude control. However, they produced very high torque. In contrast, the fuzzy controller had a good response for all three movements, with lower torque than the previous two controllers. Furthermore, the PD controller had an undesired steady-state error in yaw control, while the PID controller had acceptable performance for the three movements. Finally, the nested saturation controller had a better performance than the previous controllers, especially in altitude control, with low torque except for the yaw and roll, where it produced large torque.

In [81], three control techniques were used to enhance the control response of a fixed-wing UAV ultra stick. The first control technique used a Proportional–Integral–Derivative PID controller designed and tuned using a genetic algorithm optimization technique. The second technique was tuned by a Adaptive PID Fuzzy controller. Finally, the third control technique was based on a PID controller tuned using Optimal Local Control (LOC). The results showed that the PID controller tuned using LOC had better noise attenuation and wind disturbance rejection than adaptive fuzzy tuned PID and genetically tuned PID in both longitudinal and lateral channels.

### 4.2. Muli-Rotor UAV Controllers

In [82], three controllers were applied to control a quadcopter UAV's altitude. The first controller was the Linear Quadratic Regulator (LQR) controller. It was tuned to minimize the cost function to be a linear function. The second controller used an Integral Time Absolute Error (ITAE)-tuned PID, while the third controller used an LQR-tuned PID. The results showed that the PID controller tuned by an LQR was the fastest, with no overshoot value in the vertical speed response, while in the vertical position response, the PID controller tuned with ITAE showed a faster response than the other two controllers with less than 10 percent overshoot and 0.25 s settling time.

In [83], a novel model-based reinforcement learning algorithm was developed for high-level control for autonomous navigation of a quadcopter UAV to find an efficient route when it is constrained in battery life. The controller, called the agent, in this case, learned how to modify its response to obtain the maximum cumulative reward value over time. The model-based agent consisted of three main parts: action selection, which was responsible for direct interaction with the environment and took actions with the required rate, and model learning, which learned the surrounding environment. Moreover, it built a model for use in decision trees, and planning was performed by simulating paths starting from the current state to the maximum search depth. Finally, it updated the function with the most recent model of the environment. The results of simulations of the UAV in the ROS Gazebo environment showed that the agent can learn good behavior in few iterations, perform actions in real-time, and outperform Q-learning-based methods.

## 5. Applications of UAVS

UAVs can be used in military and civil applications. In addition, they can be deployed for indoor and outdoor applications due to their low maintenance cost, high mobility, and their ability to stay airborne. UAVs are utilized for different applications, including wireless networks, remote sensing, planetary exploration, search and rescue operations, the delivery of goods, surveillance, and precision agriculture. Some of these applications are discussed here.

### 5.1. Reconnaissance and Surveillance

UAVs can be used for intelligence, surveillance, target acquisition, and reconnaissance (ISTAR), they gather real-time information with full motion video (FMV), and are gaining popularity for large area searches and multi-intelligence capabilities. PED (processing, exploitation, and dissemination) continues to be an important subject, highlighting the need for compatibility [15].

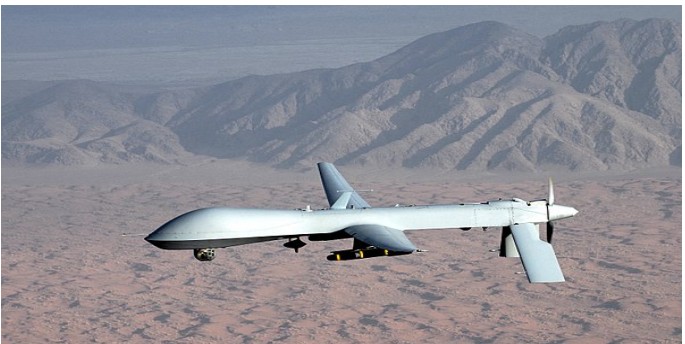

**Figure 9.** RQ-1 Predator reconnaissance and surveillance UAV [84].

As illustrated in Figure 9, the Predator can cover a large area with high hovering endurance while not exposing allied pilots to enemy fire or capture. The system acquires enemy intelligence, find targets, and fuel hostile airspace without endangering the operators' lives. UAVs can be used to examine impacted regions for chemical, biological, radiological, nuclear, or explosive materials hazards [26].

### 5.2. Combat

Unmanned combat aerial vehicles (UCAVs) should have high manoeuvrability and be capable of air to air combat, while providing precision weapon delivery to surface targets. Attack/combat UAVs have higher cruise speeds compared with other UAVs, but usually have shorter endurance [15,26]. Furthermore, Boeing continued the development of their X-45C UCAV demonstrator, shown in Figure 10, through self-funding and collaboration with NASA, proving flight capabilities in 2011.

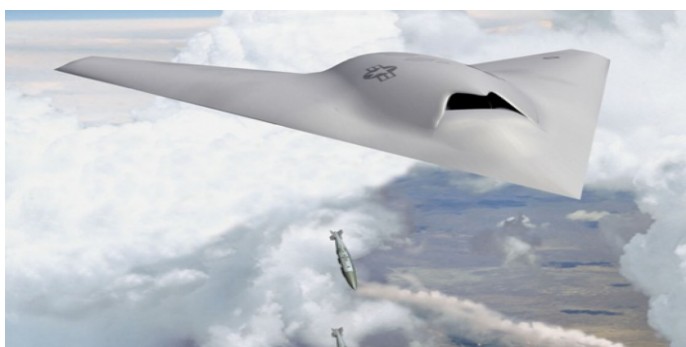

**Figure 10.** X-45C Boeing UCAV [85].

### 5.3. Humanitarian Demining

Landmines represent a major threat. Records show that there are still more than 100 million mines in the ground [86]. Detecting these landmines is a very dangerous process. Therefore, UAVs can be used for the demining process, as they have the ability to cover a wider area of any terrain in less time compared with unmanned ground vehicles (UGVs) [87] as it can be seen in Figure 11. UAVs can be equipped with various sensors to assist in the process of landmine detection; in [88] the authors used a multispectrum camera and a thermal camera to help in the detection of PFM-1 scatterable plastic landmines by applying a computer vision algorithm to automate the process of detecting and localization. In [89], a was UAV equipped with a ground penetrating radar (GPR) system to detect the landmines. Ref. [90] introduced a UAV/UGV system that employed a UGV for the detection portion, while the UAV can assist the UGV and follow it using an image processing algorithm. This system is considered, as a turning point as most current research only utilize UAVs.

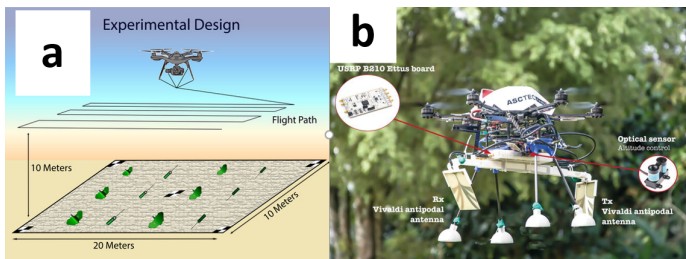

**Figure 11.** UAVs in humanitarian demining, (**a**) UAV for PFM1 plastic landmine detection using thermal infrared sensor [91], (**b**) UAV for landmine detection using ground penetrating radar (GPR) [92].

### 5.4. Public Safety

The real-time detection of air contaminants is an essential issue in air pollution hazard assessment which poses a potential threat to the environment [93,94]. It is critical to use the data acquired from pollutant concentration monitoring to locate the source. By using static sensor monitoring systems, the sensors are usually not densely spread enough to collect high-quality data [95–97]. Therefore, recent studies [98,99] have attempted to use UAVs as a monitoring platform to collect high-quality air concentration measurements of contaminants as shown in Figure 12.

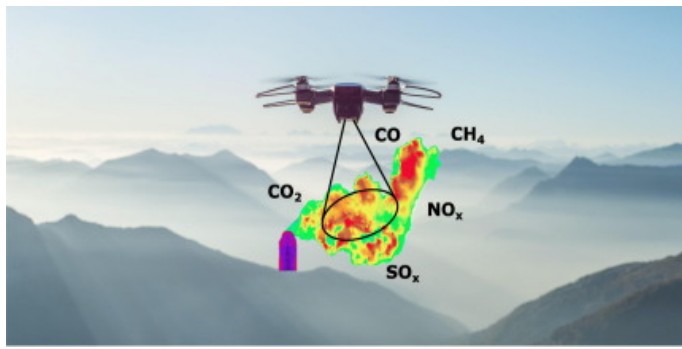

**Figure 12.** UAVs for public safety [100].

In [101], a quadcopter UAV was used as a monitoring system with a Raspberry pi micro-computer to receive the measurements from gas sensors mounted on the drone using a serial protocol. The gas monitoring system supported $O_2$, $CO_2$, $O_3$, $H_2S$, $SO_2$, CO, $NH_3$, $NO_2$, $C_2H_4$, benzene, methylbenzene, $PM_{2.5}$, and $PM_{10}$. Numerical and real experiments were conducted in an industry park in Shanghai. $SO_2$ emissions were investigated by measuring the $SO_2$ concentrations; as a result, five possible emission sources were located

in this area. The same process were employed to discover other gases contaminates. By applying this concept, the collected data were less sensitive to errors in meteorological data and concentration measurements than traditional source estimation methods.

### 5.5. Construction

UAVs are largely employed in the construction industry; they serve various aspects for a construction project, from the project planning process by aerial mapping for the site [102], to the the actual construction of buildings [103,104]. Moreover, UAVs are used for monitoring the workflow of a construction site [105,106] and inspecting buildings for maintenance [107], faults and protect them from unanticipated failures [108]. As shown in Figure 13 the UAVs could supply construction investors with comprehensive, accurate, and exact geographical data. Distributed sensors over a building measure the data then collected by a UAV and then analysed with engineering tools, and it can also be superimposed on building layouts.

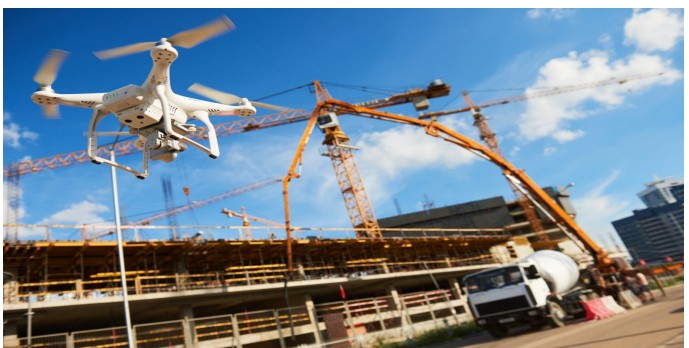

**Figure 13.** UAVs in construction industry [109].

### 5.6. Planetary Exploration

Space agencies such as NASA have shown a massive interest in designing and fabricating UAVs for planetary exploration missions, as shown in Figure 14. UAVs can be designed to fly and perform missions in different space environments, such as on Mars and Venus. Employing UAVs for planetary exploration is beneficial, as UAVs can map larger areas than rovers with higher resolutions than current orbiters [110]. However, it should be noted that the design and fabrication of space UAVs should be carried out according to the environment of the planet [14].

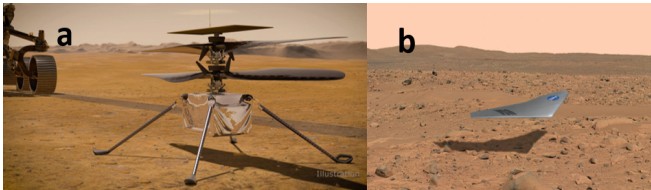

**Figure 14.** UAVs in planetary exploration, (**a**) Mars Helicopter [111], (**b**) PrandtlM [112].

### 5.7. Search and Rescue

Search and rescue operations (SAR) is a primary use-case of UAVs due to the low cost and reduced human risk in such operations. UAVs are considered to have great potential in operations such as public safety, search and rescue, and disaster management and recovery. They can assist in rescue and recovery operations needed after disasters such as earthquakes, floods, or terrorist threats, affecting critical infrastructures such as water and power utilities, transportation, and other systems, as seen in Figure 15. In addition to their ability to deliver medical supplies and emergency equipment to inaccessible regions, UAVs can be used to speed up the search in rescue operations in different dangerous situations, such as avalanches and wildfires. The process of SAR operation begins by

scanning the targeted area with a single or multi UAVs that are equipped with onboard sensors for imaging, vision, or thermal cameras. Videos/images are sent to the ground control system in real time to be analyzed. UAVs can help in SAR missions in their ability to take high-resolution images and videos using the onboard mounted cameras. They can be performed autonomously, and they can act as aerial base stations to provide services after complete communications infrastructure damage. Quad-rotor UAVs are the most commonly used in SAR missions [113].

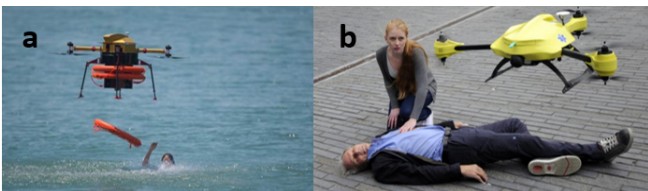

**Figure 15.** UAVs in search and rescue operations; (**a**) life-saving drone S-200 [114]; (**b**) ambulance drone [115].

*5.8. Remote Sensing*

UAVs are widely used in remote sensing applications. They collect data from ground sensors and forward the data that have been collected to a ground base station, as shown in Figure 11a. UAVs can be equipped with various sensors and used as an aerial sensor network to monitor the environment and emergency response. In addition, various applications use UAVs for remote sensing, including crop monitoring, water quality monitoring, tree species, disease detection, and mine detection, as shown in Figure 11b. Furthermore, the UAVs can help in firefighting by lowering the hazard of assessing fire disasters as presented in Figure 16. The firefighter UAVs might be used as scouts, carrying cameras equipped with thermal imaging technology to assist in rescue operations.

There are two types of remote sensing systems; active remote sensing systems include LiDAR and radar, and passive remote sensing systems include accelerometer and spectrometer. Fixed-wing UAVs and VTOL quad-rotor UAVs are usually used in remote sensing applications [113].

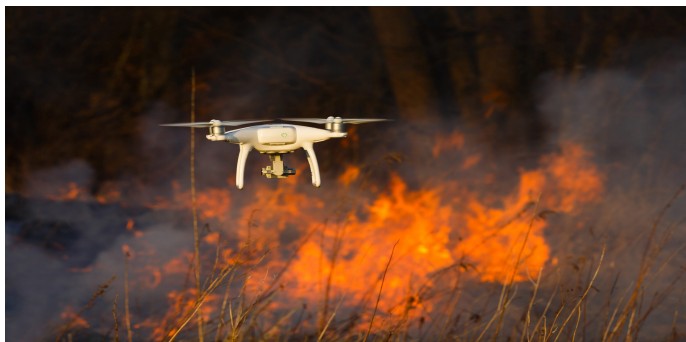

**Figure 16.** UAVs for remote sensing.

Additionally, the examination and inspection of high transmission power lines is a dangerous operation if performed by humans. Therefore, UAVs can be used for this particular task with higher safety and accuracy. UAVs can also be used for surveillance applications, including border surveillance, as they provide much broader coverage than available approaches for border surveillance. Environmental protection using UAVs, such as tracking wildfire, monitoring climate change, and the investigation of natural disasters, has increased due to the ease of implementing UAVs with various sensors such as cameras, thermal cameras, temperature sensors, and chemical sensors [116].

### 5.9. Wireless Communication

Cellular-connected UAVs are a new technology. These UAVs can cover a wide-ranging area, with better performance than traditional ground to UAV communications, due to the ease of monitoring and managing the UAVs with lower costs [117]. These UAVs can act as an aerial base station when the cellular network is damaged [118]. Furthermore, UAVs can be used to widen the ground base stations to provide better coverage with high data rates for the users, as shown in Figure 17. UAVs have other use cases in wireless communication. They can be used as gateway nodes that can be connected to a communication infrastructure or internet. The communication of UAVs with satellites is necessary to structure an integrated space–air–ground network that is able to provide high data rates anywhere, anytime, and works towards seamless wide-area coverage.

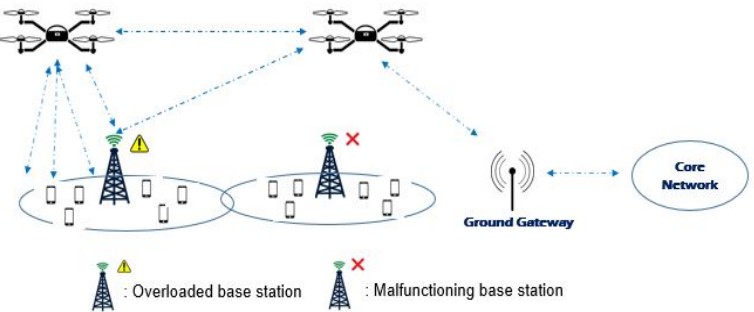

**Figure 17.** UAVs providing wireless coverage.

### 5.10. Delivery

Recently, many companies such as Amazon [119], and DHL [120] have been interested in using UAVs to carry packages and deliver them to specific locations. Amazon Prime Air developed a service that uses UAVs to deliver individual packages to their customers within 30 minutes delivery and with orders up to 2.25 kg as shown in Figure 18. Delivery UAVs can utilize vertical take-off and landing and follow a specific path to reach the desired location for delivery. UAVs can be used to deliver several things, including weapons, food, packages, and even medicines, blood samples, and organs to unreachable places, as they are faster than conventional delivery methods [113]. In addition, UAVs can be used to deliver organs for transplant surgeries. In 2019, the University of Maryland Medical Center used a custom-designed UAV to deliver a kidney to surgeons for successful transplantation in a patient with kidney failure. As a result, for the first time, it was recognized that UAVs can deliver human organs for transplantation [121].

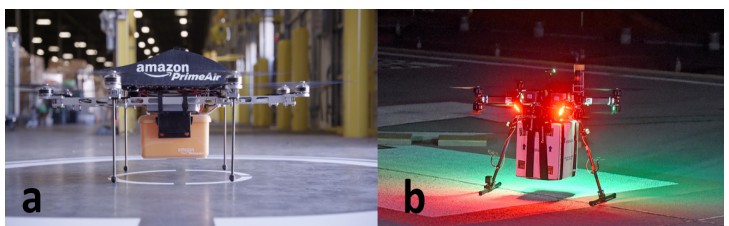

**Figure 18.** UAVs in delivery: (**a**) Amazon Prime Air [119]; (**b**) a UAV used to deliver a kidney for transplantation surgery [121].

### 5.11. Precision Agriculture

Smart agriculture uses UAVs to complete several tasks, including crop management, weed/disease detection, pesticide spraying, and irrigation, as shown in Figure 19 [122]. In contrast, they are low-cost and time-saving compared with traditional manned aircraft. In addition, UAVs in agriculture improve crop yields, increase profit, and productivity, as they provide real-time data and high-resolution images for a crop to help farmers and

stakeholders access the gathered data of the crop remotely through UAVs from cloud-based platforms [10].

In [123], several applications were discussed regarding using UAVs in precision agriculture, including monitoring crops using various sensors such as thermal, multispectral and hyperspectral cameras. Spraying systems were also covered in different aspects, including the selection of appropriate pesticides or fertilizers and the amount to be sprayed according to the plant's possible diseases and pests.

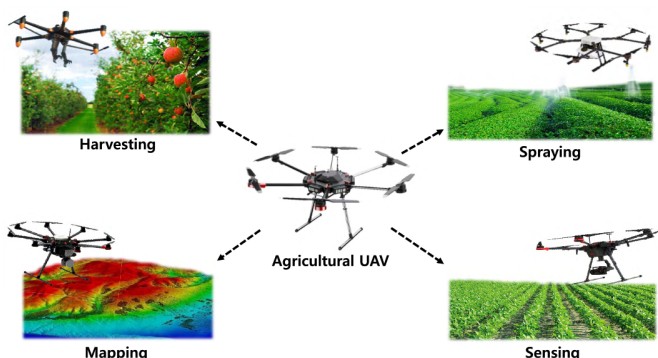

**Figure 19.** UAVs in agriculture [122].

## 6. Challenges, Limitations, and Recommendations

Despite several UAV applications, which have been discussed in the previous section, different challenges face UAVs and prevent them from achieving the required missions with high performance. In this section, we shed light on the main challenges that affect the performance of the UAVs, including battery life, collision avoidance, and security. Furthermore, their limitations and some recommendations are proposed to achieve an acceptable performance.

### 6.1. Battery Challenges

The battery capacity of a UAV is considered to be a crucial factor that affects the time needed for missions. When the battery capacity increases, the weight of the battery increases. Therefore, the UAV consumes more energy in missions. Figure 20 shows several solutions that have been presented in the literature are discussed in further detail.

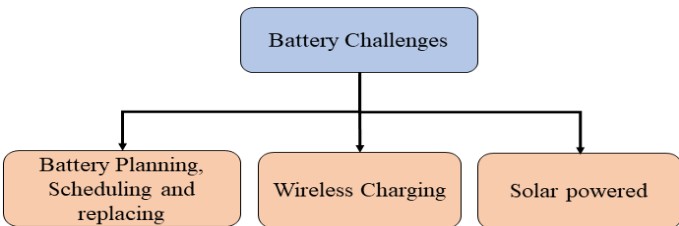

**Figure 20.** UAV battery challenges.

1.  Battery management involves the planning, scheduling, and replacement of a battery to accomplish required missions. In [124], a battery charge prediction model for UAVs was proposed to predict the end of the battery charge based on the particle filter algorithm. The discharge curve for the Li-Po battery was employed to tune the filter. In order to minimize the drop in battery life, battery assignment and scheduling for UAVs can be carried out. A heuristic algorithm can be used, as in [125], to solve the battery assignment and scheduling problems based on the time between charges and the discharge time. Swapping battery systems is a great solution for autonomous UAVs. These systems consist of a landing platform, battery charger, battery storage, and microcontroller [126]. The hot-swap system is similar to these, which powers the UAV continuously during the battery swapping process as the UAV is connected to an

external power source. The swapping process is performed to prevent data loss, and then, the battery swapping action is performed. After that, the UAV is disconnected from the power supply to continue its mission [127].

2.  Some researchers have presented wireless charging in the literature. One of the proposed studies discussed recharging UAVs from power lines during the inspection process [128]. Another proposed solution is to provide an automatic charging station system for UAVs distributed on the UAV given path. This station system consists of four parts: a solar panel, a wireless charging pad, battery, and a power converter [129].

3.  Solar-powered UAVs provide high altitudes and long endurance flights, while solar power acts as the primary power source for the propulsion system. At the same time, the batteries are considered as the secondary source to be used at night and in conditions of the total absence of direct sunlight [130].

### 6.2. Collision Avoidance Challenges

Collision avoidance for UAVs is a great challenge, as it is essential to avoid accidents with any obstacle. Obstacles can be moving or stationary, and UAVs may crash with them. Therefore UAV collision avoidance techniques have emerged recently, which include two approaches: the geometric approach, which considers geometric analysis to avoid the collision, the path planning approach, which uses a geometric technique to find a path without collision during the flight [131], and the vision-based approach, which captures images from cameras mounted on the UAV to avoid the collision challenge with the help of image processing operations which can be performed onboard.

The collision avoidance system automatically detects and avoids collisions and quickly alerts the UAS operator, who may decide to assume control or propose prompt action to resolve the situation. The flow and sequence of sensing, detecting, and avoiding (SDA) are presented in Figure 21.

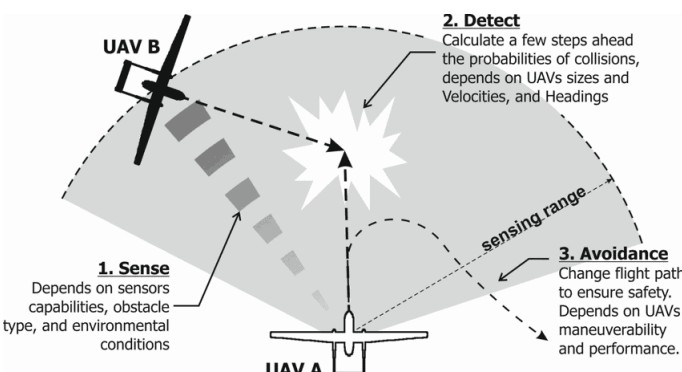

**Figure 21.** Cooperative autonomous collision avoidance system [132].

In [133], for the drone route, the writers created a route management system (RMS). By designating particular routes for drones, this approach can improve drone safety at airports. A collision avoidance system should be built for this system in order to avoid any possible collisions between the drones by maintaining a safe distance between them. That safe interval can be calculated as follows:

$$S_{int} = T_{res} + B_{dis} + L_{hc}.2 \tag{7}$$

where $S_{int}$ is the safe interval, $T_{res}$ is the distance traveled by the drone from the time it detects an obstruction to the time it hits the emergency braking, $B_{dis}$ is the distance traveled by the drone after triggering the emergence braking, and $L_{hc}$ the distance between the drone and the monorail's ground mobile platform. A safe distance may be calculated using this formula.

### *6.3. Security Challenges*

The security of the UAV is essential in several components of the UAVs. They can be attacked by intruders, which in return create cybersecurity challenges for the UAV systems, as shown in Figure 22. Several attacks can target the communication links between different equipment in the UAV system. In addition, UAVs themselves can face direct attacks, which can cause dangerous damages to the UAV system, such as signal spoofing and hacking. The GCS attacks are serious because they enable the intruders to send commands to the devices and collect all the data from the UAV, which can be fatal and may cause failure for the UAV [134].

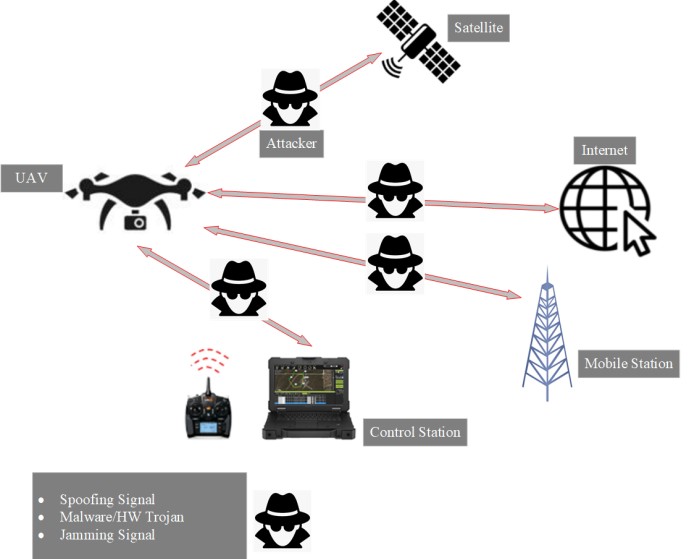

**Figure 22.** Security issues in UAVs.

### *6.4. Other Challenges*

With the legalization of UAVs in the US, the FAA does not currently permit swarms of autonomous UAVs for commercial uses. In addition, environmental conditions are another challenge to UAVs, as they may result in deviations in pre-determined paths or may crash the UAV itself.

### 7. Future Research Trends

Despite efforts to propose acceptable solutions to address the UAV challenges discussed in the previous section, a significant amount of unsolved difficulties require new effective solutions. In this section, we highlight new opportunities for the UAV systems, including security and privacy, battery charging, machine learning, and other interesting future research topics and directions.

### *7.1. Swarm Uav Systems*

In swarm UAV systems, a set of UAVs work together to achieve a specific goal. Each UAV has a small mission which is a part of a bigger mission; this concept was mainly developed by the military for reconnaissance and surveillance applications. Civilian applications also adopted the same concept for several applications, including precision agriculture and SAR operations [135].

### *7.2. Machine Learning and Deep Learning*

Machine learning and deep learning algorithms have recently received a high level of support in different applications related to UAVs, such as resource allocation, obstacle avoidance, tracking, path planning, and battery scheduling. The development of more accurate algorithms and improvement of onboard computational power will lead to the

design of nano UAVs that are much smaller, more lightweight, and smarter than available UAV models to achieve the required mission accurately and without the risk of collision. Furthermore, the availability of accurate data can facilitate UAVs to perform accurate control and path planning and vision tasks [136].

In [137], several applications of deep learning in UAVs were reviewed, such as feature extraction using UAVs. By implementing different cameras on the UAV, various types of images can be taken for further processing. UAV planning, including path motion, navigation, and manipulation planning were presented, as the UAV navigates the environment in order to find a suitable path. Another application is the motion control of UAVs based on deep learning.

### 7.3. Security and Privacy

The security and safety of UAVs are essential parameters due to their wireless connectivity and limited computational capabilities. They receive potential threats from intruder attacks which can disrupt the privacy and confidentiality of the collected data. It might be stolen or replaced. Therefore, new onboard techniques are required to ensure privacy on the mission. Recent techniques, such as blockchain and physical layer security, require further research and improvement to achieve a required security level with the required quality and reliability [138].

### 7.4. Trajectory and Path Planning

Tracking and path planning techniques should be improved to optimize the mission path of UAVs while minimizing the energy consumption of flights with collision avoidance. Recent work on the topic of tracking is mainly based on heuristic algorithms, while path planning of complex paths to avoid obstacles and find the shortest path to consume low energy can be carried out by using a multi-objective optimization algorithm [138].

### 7.5. Energy Charging

Recent developments in battery technologies, including enhanced lithium–ion batteries and hydrogen fuel cells and green energy sources such as solar energy, have been used recently to extend flight times. However, energy collecting efficiency is low due to random energy arrivals. Novel energy-delivering technologies, such as energy beamforming through multiantenna techniques and distributed multipoint WPT, can enhance the charging efficiency [139].

### 7.6. Optical Communication

Optical Wireless Communications (OWCs) have proved their efficiency in 4G, 5G, and beyond 5G mobile networks. They are widely adopted in UAV communication, and they are expected to be used in the 6G mobile network. However, several challenges facing this technology need to be addressed, including the high blockage probability of the signals, the power consumption, and weather conditions [140].

## 8. Conclusions

The use of UAVs has become widespread in many fields, including military and civil fields. Different classifications of UAVs were proposed. They can be classified into four classes according to size: ultra-small UAVs, including MAVs and NAVs, small UAVs, mid, and large UAVs. Classifications according to the range, endurance, maximum altitude, and weight were also represented and discussed in detail. Diversity in configurations of the UAVs were investigated, as UAVs can be classified into four main categories: their take-off and landing method, HTOL, VTOL, hybrid, and unconventional UAVs; each one was described in detail. The design process of the UAVs was demonstrated for the general design process. It consists of three stages: conceptual design, preliminary design, and detailed design. Furthermore, the available FEA analysis methods were described: structural analysis, vibration analysis, and computational fluid dynamics (CFD). The

control system architecture and goals were discussed, as well as the various control laws that can be used to control UAVs, such as PID, LQR, Backstepping, gain-scheduling, and NDI. Several applications employ UAVs, such as planetary exploration, search and rescue operations, remote sensing, surveillance, real-time monitoring, environmental protection, wireless communication, delivery, and agriculture. Finally, challenges, limitations, and recommendations were also investigated.

**Author Contributions:** The authors contributed to the manuscript equally. All authors have read and agreed to the published version of the manuscript.

**Funding:** This paper is based upon work supported by the Science, Technology and Innovation Funding Authority (STIFA) of Egypt, under grant no. (38121).

**Institutional Review Board Statement:** Not applicable: the study did not involve humans or animals.

**Informed Consent Statement:** Not applicable: the study did not involve humans or animals.

**Data Availability Statement:** This study did not report any data.

**Conflicts of Interest:** The authors declare that there is no conflict of interest. The authors whose names are listed immediately below have seen and agree with the manuscript's contents, and there is no financial interest to report. We certify that the submission is original work and is not under review at any other publication.

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
