# Peer review of "A Detailed Survey and Future Directions of Unmanned Aerial Vehicles (UAVs) with Potential Applications"

_aerospace, doi:10.3390/aerospace8120363_

Round 1
Reviewer 1 Report
The style of writing changes throughout the text.
Units of measurement used change in different parts between meter and ft.
Various types of UAV technologies, applications, etc. are mentioned, however no bibliographical references are presented.
Design methodologies for (fixed-wing UAVs, multirotor UAV) are mentioned, however, there is no minimum mathematical formalization in the design of such UAVs.
Finally, the abstract talks about possible future applications of UAVs (such as trajectory planning, control, etc. ), however, the concepts are very lightly addressed and no details are given.
Author Response
Dear Reviewer,
We would like to thank the reviewer for taking the time and effort necessary to review the manuscript. We sincerely appreciate all the valuable comments and suggestions which helped us to improve the manuscript’s quality. We hope that the revised paper meets the reviewer’s expectations, and thus you can accept it for publication.
Best regards,
Nourhan Elmeseiry, Nancy Alshaer and Tawfik Ismail

Reviewer 2 Report
The research progress and future trend of UAVs are investigated in detail. Some suggestions are given as follows:
1. Some common sense of UAVs can be curtailed, such as the history of UAVs.
2. The font size in Fig. 7 can be enlarged.
3. For the challenges, why do the authors select the following aspects to illustrate: battery charging, collision avoidance, and security? What is the logical relationship among these four aspects?
4. It seems that the applications of military/police UAVs and new-energy UAVs are neglected.
Author Response

(The authors gave the same response as above.)

Reviewer 3 Report
The authors made an extensive overview in the manuscript of the development of UAV technology, this is the strong point of the work.
But at the same time, the authors did not highlight a number of important points in the manuscript:
Section 2.2. and Table 2 does not take into account such a type of UAV as a tethered drone (one of the examples is https://elistair.com/). Tethered drones receive energy through the holding cable and for this reason can remain in the air indefinitely.
Section 5 does not take into account such a promising type of UAV application as ensuring public safety and ensuring the safety of technological processes at industrial enterprises, especially at potentially hazardous ones (chemical production, etc.).
The section does not take into account the drone collision avoidance method proposed in the article https://link.springer.com/article/10.1007/s12198-020-00225-z
Author Response

(The authors gave the same response as above.)

Round 2
Reviewer 3 Report
The work carried out by the authors combined a significant overview of the development of UAV technology, it will certainly be useful to researchers and students related to the field of aerospace technology.
Author Response
We would like to thank the reviewer for considerate reviews and efforts to improve the manuscript’s quality. We sincerely appreciate all valuable comments and suggestions, which helped us to improve the quality of the article